# Facebook and Suicidal Behaviour: User Experiences of Suicide Notes, Live-Streaming, Grieving and Preventive Strategies—A Scoping Review

**DOI:** 10.3390/ijerph192013001

**Published:** 2022-10-11

**Authors:** Sheikh Shoib, Miyuru Chandradasa, Mahsa Nahidi, Tan Weiling Amanda, Sonia Khan, Fahimeh Saeed, Sarya Swed, Marianna Mazza, Marco Di Nicola, Giovanni Martinotti, Massimo Di Giannantonio, Aishatu Yusha’u Armiya’u, Domenico De Berardis

**Affiliations:** 1Department of Psychiatry, Jawahar Lal Nehru Memorial Hospital, Srinagar 190003, India; 2Department of Psychiatry, University of Kelaniya, Kelaniya 11300, Sri Lanka; 3Psychiatry and Behavioral Sciences Research Center, Mashhad University of Medical Sciences, Mashhad 9177948564, Iran; 4Alice Lee Centre for Nursing Studies, Yong Loo Lin School of Medicine, National University of Singapore, Singapore 117597, Singapore; 5Frontier Medical and Dental College, Abbottabad 22010, Pakistan; 6Department of Psychiatry, Psychosis Research Center, University of Social Welfare and Rehabilitation Sciences, Tehran 1985713834, Iran; 7Faculty of Medicine, Aleppo University, Aleppo 12212, Syria; 8Department of Geriatrics, Neuroscience and Orthopedics, Institute of Psychiatry and Psychology, Fondazione Policlinico Universitario A. Gemelli, IRCCS, 00168 Rome, Italy; 9Department of Psychiatry, Fondazione Policlinico Universitario Agostino Gemelli, IRCCS, 00168 Rome, Italy; 10Department of Neuroscience and Imaging, University “G. d’Annunzio”, 66100 Chieti, Italy; 11Department of Psychiatry, College of Medical Sciences, Abubakar Tafawa Balewa University, Bauchi PMB 0248, Nigeria; 12Department of Mental Health, ASL 4, 64100 Teramo, Italy

**Keywords:** suicide attempt, behaviour, prevention, Facebook

## Abstract

Background: Facebook represents a new dimension for global information sharing. Suicidal behaviours and attempts are increasingly reported on Facebook. This scoping review explores the various aspects of suicidal behaviours associated with Facebook, discussing the challenges and preventive measures. Methods: PubMed, Google Scholar, and Scopus were searched for related articles published in English up to October 2021, using different combinations of “Facebook” and “suicide”. A group of experts comprising consultant psychiatrists screened the records and read the full-text articles to extract relevant data. Twenty-eight articles were chosen as relevant and included in the review under four selected themes. Results: Facebook impacts on suicidal behaviours in different aspects. Announcing suicides through sharing notes or personal information may lead to the prediction of suicide but be harmful to the online audience. Live-streaming videos of suicide is another aspect that questions Facebook’s ability to monitor shared contents that can negatively affect the audience. A positive impact is helping bereaved families to share feelings and seek support online, commemorating the lost person by sharing their photos. Moreover, it can provide real-world details of everyday user behaviours, which help predict suicide risk, primarily through novel machine-learning techniques, and provide early warning and valuable help to prevent it. It can also provide a timeline of the user’s activities and state of mind before suicide. Conclusions: Social media can detect suicidal tendencies, support those seeking help, comfort family and friends with their grief, and provide insights via timelining the users’ activities leading to their suicide. One of the limitations was the lack of quantitative studies evaluating preventative efforts on Facebook. The creators’ commitment and the users’ social responsibility will be required to create a mentally healthy Facebook environment.

## 1. Introduction: Suicides and Social Media

Globally, 700,000 people die yearly by suicide, and 77% of these suicides occur in low and middle-income countries [1]. Suicides are frequently under-reported for various societal, economic, and political reasons; therefore, the actual number of suicides is believed to be significantly higher [2]. Early cases have been documented wherein users took to social media to post suicide notes, announce, or even broadcast their suicide attempts [3]. Social connectedness, school support, and family relationships are major protective factors against suicide. Further, the use of distractions, problem-solving skills, and high self-esteem reduced the risk of developing suicidal ideation [4]. Some risk factors in young people for suicide include lack of social support, imprisonment, poor life skills, family history, diagnosed mental disorders, adverse life events, abuse in childhood, academic stress, use of alcohol, and cyberbullying [5].

Today, social media has become a mainstay of communication, and out of the diverse options available, Facebook is the largest known platform, with close to three billion users [6]. Social media broadens the scope and content of human communication and allows for free expression and selective representation of undesirable behaviour [7]. An emerging trend in the last few years is announcing suicide on Facebook [8]. It is observed that young people who self-harm use the Internet frequently to express their distress, and there is a rising trend of people dying by suicide after posting on social media, which is found to have assortative patterns [9]. Certain users post their intent publicly on social media and then die by suicide, and a number of such cases have been reported [6].

According to studies, expressing suicidal intent via social media platforms might be seen as an unconventional means of seeking help, and this has encouraged researchers to look into harnessing the powers of social media to prevent suicides [10]. It is observed that users are keen to be helpful; however, they lack the knowledge required. Efforts focused on empowering social media users, such as forming rescue or support groups, would make for a convivial and welcoming online world [11,12,13]. Prevention of suicides by monitoring social media posts and analysing online behaviour would be possible [14]. Artificial intelligence-driven suicide prediction methods are tested and used that could improve the capacity to identify those at risk of self-harm and suicide, and, hopefully, could save lives [15].

Although the associations between suicidal behaviour and social media can be investigated in several psychosocial aspects, few studies have addressed this issue to consolidate the relevant evidence available in the literature. There is currently no report on the percentage of Facebook-associated suicides that lead to death. The extent of the detrimental effects that social media can have in this regard, and their possible contributions to predicting and preventing suicidal behaviours remain controversial. We aimed to identify factors related to announcing suicide on Facebook, live-streaming of suicidal behaviour, grieving suicide, and preventing suicides to understand the socio-cultural implications and the role of the audience and how one can prevent such events. Out of all the social media platforms, we focused on Facebook, which has the largest number of followers and a unique online culture [16,17].

## 2. Methods

We searched Google Scholar, Scopus, and PubMed up to October 2021 from the beginning of data for articles related to announcing suicide on Facebook, live-streaming of suicidal behaviour, grieving suicide, and preventing suicides. The selected themes were decided by a panel of experts comprising consultant psychiatrists (authors) who have published on mental-health aspects of social media use. The search strategy used on PubMed was ‘Facebook suicide [Title/Abstract]) OR (Facebook suicidal [Title/Abstract]) AND (English Language), and it yielded seventy-one initial results. We used the search term “Facebook suicide” on Google Scholar, leading to 246 article findings. We searched article title, abstract, and keywords as “Facebook suicide” on Scopus, leading to 175 articles. A relevant article was only selected if the publication provided information about an association between suicidal behaviour and Facebook use in the selected themes. A group of consultant psychiatrists went through all of the articles, and data were extracted from each original article. Only English language peer-reviewed studies were included in the review. We also reviewed the reference lists of included articles for additional publications. Abstracts, unpublished research, reviews, and duplicates were excluded. All search results were perused on the relevance to the topics announcing suicide on Facebook, live-streaming of suicidal behaviour, grieving suicide, and preventing suicides. The articles not reporting data on the above themes were excluded after two consultant psychiatrists studied each one. Consultant psychiatrists as experts identified key points relevant to each theme in the study. A key point was defined as a finding of the selected study that is relevant for identifying and preventing suicides concerning Facebook use.

## 3. Results

Figure 1 shows the study selection process. We have summarised the significant findings of the most relevant studies that were retrieved by our search of the literature into four distinctive categories: announcement of suicide decision on Facebook (Table 1); video-streaming of the act of suicide on Facebook (Table 2); using Facebook for grieving after a suicide (Table 3); and using Facebook for prevention of suicide (Table 4). The tables include key points relevant to the topics and themes selected by the consultant psychiatrists. As indicated in Table 1, the announcement of suicide on Facebook can be made through sharing suicide notes, personal information, or preceding life events leading to suicide, which can be helpful in the prediction of suicide. Facebook also provides a large audience with whom the suicidal behaviours can be shared, which can negatively affect them.

Table 2 summarises the reported data on cases where Facebook shared the live-streaming of videos of suicide online. The main concerns were the ability of Facebook to monitor the content of these videos and how it can prevent their spread, which can negatively affect the audience.

One of the major advantages of Facebook is the forum it provides for grieving family members or friends, through which they can share feelings and obtain support from other users online, as well as commemorating the dead person by sharing their photos or posts (Table 3).

Table 4 summarises the studies on another advantageous use of Facebook: suicide prevention. There have been different suggestions and actions in this regard, such as artificial intelligence algorithms that can predict and help prevent suicides, online support groups where people can share their distress, reduce the risk of suicide, and help other users who commiserate with the person sharing their hardships and feelings.

## 4. Discussion

Ref. [1], we explored the association between suicidal behaviour with Facebook and selected 28 studies as relevant out of 492 search findings. The review focused on four areas: announcing suicide; live-streaming; grieving suicide; and prevention strategies.

More people sharing personal information openly on Facebook allows them a chance to reach out to others even without knowing them. There are several reports of reported suicides where a suicide note was shared on Facebook beforehand [3,6,18,20]. In emotionally distressful situations, it may be possible to offer support and prevent a potential suicide [24]. By analysing public posts and suicide notes, social media platforms should identify users at an elevated risk and provide online support [3,6,18]. Suicide notes found on Facebook were associated with a younger age, psychoactive substance use, and previous non-suicidal self-injury in the individual [20].

Further, young people could express and share their suicidal thoughts on social media more conveniently than with their families or professionals [8]. An unrestricted reach for self-expression was provided by Facebook and encouraged people in distress to normalise self-harming behaviour, and some even competed with others to injure themselves more seriously [21,22]. Apart from the opportunity to express their distress, repetitive negative thinking, which is likely to be present in many in emotional distress, and addictive Facebook use are associated with suicidal ideation and suicidal behaviour. This may have led to a social–emotional environment invigorating potential harm directed at self [40].

Few studies have investigated the links between social media use, personality disorders, and traits [42,43]. For example, having more Facebook friends has been associated with mania, histrionic, and narcissistic personality disorder symptoms, and fewer symptoms of dysthymia and schizoid personality disorder [43]. Facebook users, compared to non-users, reportedly had higher self-esteem and narcissism levels [42]. These findings possibly underscore the crucial impacts of personality disorders and their traits on suicidal behaviours via social media.

People stream their suicide attempts live, and timely interventions with the support of the regional mental health teams are required to save lives [23]. At times, loved ones are left helpless by such incidents, and some have taken legal action against the social media platform [44]. Persons who live-streamed suicidal behaviour were mainly under 35 years of age, and a majority were male students resorting to hanging and poisoning to harm themselves [24]. The victims had experienced relationship conflicts and academic stressors, as seen by their most recent posts before harm, and it is shown that stress-management interventions through Facebook are beneficial for young people [45].

Memorial pages on Facebook have given a platform for loved ones to mourn the loss of a family member or even a fictional character due to suicide [25,27,29]. These pages contained long, detailed posts by loved ones and words suggesting they struggled with their losses [28]. Women experiencing a loss and grieving were more likely to be Facebook subscribers, and mental health messages would be able to offer psychological support [26]. Furthermore, research shows that many individuals use Internet forums and Facebook groups to grieve losses and find emotional support [25,27]. Despite many people seeking face-to-face professional support, they preferred informal support from online platforms such as Facebook. For many, Facebook allowed bereaved people to memorialise their loved ones and feel their ongoing virtual presence [3,18]. Individuals post on Facebook about their loss and grief to commemorate the deceased loved ones, mourn, and remember their special occasions, allowing them to vent their distress in a virtual environment [30,46].

Multi-task artificial neural network models analysing Facebook posts could predict suicide risk better than a single-task model [31]. This will facilitate building tools to predict suicide risk among users correctly and has successfully dealt with crises in inpatient settings [36]. Users on Facebook can anonymously post their distress, minimal suppression of emotional expression, and prevent interpersonal communication breakdowns [33,34]. Social media, such as Facebook, provides an opportunity to talk to others and provide emotional support. There are substantial benefits of social media in suicide prevention [35]. However, Facebook needs to adhere to suicide-reporting guidelines, as harmful content is frequently seen and should prevent the glorification of suicidal deaths [37,38]. In the future, Facebook has demonstrated efficacy and cost-efficiency in recruiting people for suicide prevention activities which could be used to develop far-reaching strategies [39]. Newer technologies, such as artificial intelligence and machine learning, will play a core role in future preventive strategies, and finding the correct balance between safety and ethics will be a significant challenge [47]. Facebook, indeed, needs an algorithm to detect and prevent suicide in action and to achieve this, machine learning classifiers need to be built, and many examples need to be fed into the system [48]. Various combinations of words impact on the classifier’s confidence, and it scores the content based on previously confirmed cases of suicidal expression. The classifier scores are inserted into a random forest learning algorithm, a machine learning type specialising in numerical data [48].

Our scoping review showed both positive and negative effects of Facebook on suicidal behaviour. It provided an audience to express distress and live-stream harmful behaviour. On the other hand, Facebook allowed loved ones to mourn virtually and share their grief with others. Further, limited studies showed that Facebook could provide opportunities to prevent suicides and function as a protective agent. As a limitation, there were only a few quantitative studies to analyse, and it was challenging to reach broader conclusions. In addition, the studies that were found were heterogeneous in methodology and approach.

Further, our methodology only focused on four selected themes and a scoping approach for the review. Most data were qualitative, and establishing causal relationships between Facebook use and suicidal behaviour is complex. Creating a psychologically healthy Facebook experience will depend on the developers’ commitment and the users’ social responsibility, and further longitudinal quantitative studies are required to investigate the association between Facebook use and suicidal risk. Lastly, we did not investigate other factors influencing the use of Facebook and suicide.

## 5. Future Clinical Implications

Our review found numerous reports of posting suicide notes on Facebook and the live-streaming of self-harm viewed by many. In addition, people were using Facebook to mourn the loss of loved ones and newer technologies are being used to detect high-risk posts early so that interventions can be offered. Whether the suicide prevention algorithm of Facebook and other social media can effectively reduce the rate of suicides is an issue that requires further evaluation in large-scale, well-designed studies [49]. The possible association between the excessive use of social media networks and a higher risk of suicidal behaviours remains controversial and needs further investigation. As stated in previous studies, the extensive use of social media as an addictive behaviour can increase the likelihood of suicidal behaviours through interactions with specific users, social groups, or online communities [50]. This is of cardinal importance in adolescents and young adults, in which imitative suicide is a major mechanism.

One important matter that deserves close attention from clinicians, social media regulators, and policymaking authorities is the ethical and moral issues that may arise when breaching the privacy of social media users by looking into and analysing their details. This may be indeed necessary for specialists who try to help the users and prevent deaths by suicide [49].

## 6. Recommendations

We recommend that future studies focus on real-world data and algorithms that provide insight into self-harm risk based on social media information. Moreover, further studies are required to shed light on the possible link between the addictive use of social media and suicidal behaviours;State authorities and policymakers are recommended to enforce strict data sharing regulations so that users’ personal information and privacy are not affected. Artificial intelligence methods should anonymise the necessary information and provide valuable information to therapists;Advertisement campaigns regarding suicide prevention on Facebook can promote the use of hotlines and mobile applications, providing dependable and practical support for people in distress;Facebook can also efficiently increase the general population’s awareness regarding mental health issues and educate the communities to prevent the stigmatisation of mental illnesses. This can reduce the rate of suicidal behaviours and deaths by suicide in the long term;An option for being under surveillance for suicide risk should be extended to all users;All social media platforms, including Facebook, should use techniques for users to report harmful websites and activities of other users;Direct and quick ways to obtain psychological support via Facebook should be available;Public health initiatives should target Facebook to promote awareness of psychological problems in schools, colleges, and other settings;Those in charge of suicide prevention and public-health outreach efforts must keep up with Facebook trends, user preferences, and relevant legal issues;Finally, using Facebook to raise public awareness and education about mental health issues is a sensible modern public health strategy to save lives.

## 7. Conclusions

Social media can detect suicidal tendencies, support those seeking help, comfort family and friends with their grief, and provide insights via timelining the users’ activities leading to their suicide. One of the limitations was the lack of quantitative studies evaluating preventative efforts on Facebook. The creators’ commitment and the users’ social responsibility will be required to create a mentally healthy Facebook environment.

## Figures and Tables

**Figure 1 ijerph-19-13001-f001:**
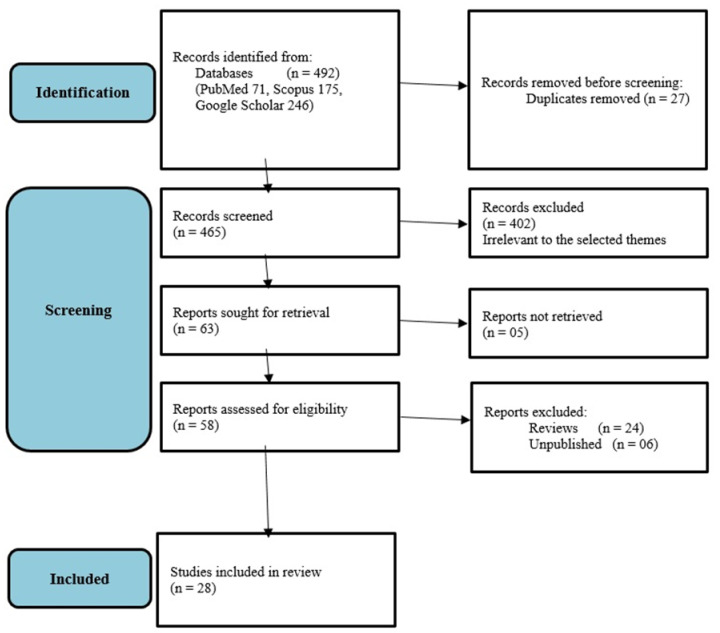
Publication selection process.

**Table 1 ijerph-19-13001-t001:** Announcing suicide on Facebook.

Authors	Article Type	Report	Key Points
Behera et al., 2020 [6] (NO_PRINTED_FORM)	Case presentation and discussion	A 32-year-old male who died by suicide was discovered hanging at his residence. The investigation found that he had uploaded a suicide note to Facebook. Images of the ligature and multiple messages regarding his intention were mentioned earlier on his account, and he had requested his online friends to support his family.	People share personal information more openly on Facebook. Posting about their suicide on social media gives the victim the chance to reach out to others without meeting or even knowing them. It may be possible for one of the Facebook contacts to offer support and intervene to prevent suicide.
Ahuja et al., 2014 [18]	Clinical case discussion	A patient was hospitalised after making an impulsive suicide attempt, and social media was used to identify the events leading up to the attempt. This evidence helped the patient gain more insight and agree to participate actively in treatment.	The timeline of social media posts can speculate the events before a person’s suicide attempt enabling the identification of triggers. Social media platforms may be valuable in identifying and preventing suicide by screening users at considerable risk and providing online support.
Ruder et al., 2011 [3] (NO_PRINTED_FORM)	Case report	A case involving a suicide note on Facebook discussed potential consequences.	Suicide notes on social media may help prevent suicide as other users could intervene and extend support. The extent of copycat suicides on Facebook is unclear.
Soron, 2019 [19]	Case report	A 25-year-old female died by suicide by hanging in Bangladesh. Numerous posts on her Facebook were evidence of her deteriorating mental state and her suicidal ideation. Nevertheless, she was ridiculed and goaded by saying, “you should die”, while others thought it was funny or frivolous.	There is increasing interest in finding how Facebook can be used for suicide prevention. For suicide prevention efforts to succeed, users’ appropriate and active participation must be ensured.
Barrett et al., 2016 [20]	Cross-sectional study	Among 1435 non-fatal self-harm cases, 44 left a social media suicide note, and 71 left a paper suicide note. Clinical notes of clients presenting with self-harm to two emergency departments were searched for mentions of social media use. Risk factors were compared to clients who used paper notes.	It was observed that leaving suicide notes on social media was associated with younger age, substance use, and repeated non-suicidal self-injury. On the other hand, leaving a paper note correlated with higher suicidal intent and risk.
Islam et al., 2021 [8]	Case series	In Bangladesh, nine cases who expressed their intent on Facebook were studied. A series of reports in which victims died by suicide after sharing their distressed thoughts and emotions on Facebook posts or live-streaming. Most of the victims were adolescents and youth.	Adolescents and young adults frequently share their suicidal thoughts on social media rather than with their families or mental health professionals. Artificial intelligence and machine learning are recommended to screen potential suicidal content on social media.
Rossi and de Silva, 2020 [21]	Case report	A male patient presented to the emergency room after an intentional overdose following an altercation with his girlfriend. He had posted, “there is no coming back from this”. There were thousands of views, with some viewers encouraging him and influencing his behaviour.	Social media provides a large audience and unrestricted reach for self-expression, encouraging people at risk of self-harm and suicide. The social media environment also normalises suicide and self-injury and may encourage competition between users to injure themselves more seriously.
Kailasam and Samuels, 2015 [22]	Case presentation and discussion	In the USA, two cases were presented who left suicide notes on Facebook before an attempt.	Social media could be recruited for early intervention. High-tech monitoring methods should be administrated for high-risk patients.

**Table 2 ijerph-19-13001-t002:** Live-streaming suicide on Facebook.

Authors	Article Type	Report	Key Points
Majeed et al., 2018 [23]	Case report	A 14-year-old male read a suicide note and took a handful of pills before a live stream after a conflict with his family. A viewer called the emergency services, resulting in a successful rescue. The teen later stated that he waited till he had as many viewers as possible before attempting suicide.	Artificial intelligence is used to monitor online content for suicidal intentions. Despite this, people stream their suicide attempts live. Timely intervention and collaboration with local authorities are required to save lives.
Soron and Islam, 2020 [24]	Cross-sectional study	In Bangladesh, 19 cases who died by suicide were studied, and this was an online search to find people who died by suicide after Facebook posts or live streamed suicide.	All were under 35 years of age, and a majority were male students. Hanging and poisoning were the most frequent methods. Facebook posts and live stream videos indicated relationship problems and academic stressors.

**Table 3 ijerph-19-13001-t003:** Facebook and grieving suicide.

Authors	Article Type	Report	Key Points
Degroot and Leith, 2018 [25]	Qualitative study	About 3600 posts in a Facebook memorial group were analysed in the USA. In 2009, a memorial page was created on Facebook to mourn the suicide of a fictional character, Lawrence Kutner, from the sitcom House MD. Parasocial relationships and breakups were most of the content posted by fans.	Social media might be responsible for a blurred line between fiction and reality and present as an outlet for grief.
Feigelman et al., 2019 [26]	Cross-sectional study	In the USA, 1432 adults in the general population were studied, investigating the correlates of suicide and bereavement.	Females experiencing bereavement were more likely to be Facebook subscribers, and target messages on this platform could offer psychological support.
Bailey et al., 2017 [27]	Cross-sectional study	In Australia, 222 suicide bereavement forum users were surveyed, and help-seeking behaviours, potential benefits, and limitations were studied.	More than 90% of the participants sought face-to-face professional support. However, they preferred informal support provided by the online platform and were willing to continue this engagement.
Scourfield et al., 2020 [28]	Quantitative comparative study	A UK study using artificial intelligence to assess Facebook memorial notes related to 23 suicides and 29 road traffic accidents was conducted. Unique features of online memorials for youth suicides were explored and compared with memorials for road traffic accident deaths.	Memorial sites on Facebook contained more detailed posts and words suggesting causation and achievement. Compared to road traffic accident deaths, suicide memorial posts had more tentative words, non-fluencies, and question marks, suggesting that loved ones struggled to make sense of these losses.
Bell et al., 2015 [29]	Qualitative study	In the UK, eleven bereaved persons (20–60 years old) were analysed through interviews, and some individuals have created, maintained, and used Facebook sites to remember a loved one who died by suicide.	For many, Facebook allowed the bereaved to memorialise their loved ones and feel their ongoing virtual presence. It positively affected their mental health but created tension with others dealing with grief differently.
Dilmac, 2018 [30]	Qualitative study	An analysis of Facebook accounts dedicated to Turkish martyrs described the new funeral rituals seen on the Internet.	With the virtual tombs, memorials, and death anniversaries, the virtual world allows families to accept the Turkish soldiers’ deaths and choose to keep them ‘alive’ online.

**Table 4 ijerph-19-13001-t004:** Facebook and Suicide Prevention.

Authors	Article Type	Report	Key Points
Ophir et al., 2020 [31]	Cross-sectional study	Analysed 83,292 posts of 1002 authenticated Facebook users. Artificial Neural Network (ANN) models were used to screen suicide risk from everyday posts of social media users.	A multi-task model predicted suicide risk with higher accuracy by analysing Facebook posts than a single-task model. Tools can be developed to predict suicide risk among social media users by analysing their posts.
Robinson et al., 2021 [32]	Online cross-sectional survey	Reported an online survey (*n* = 48) and two workshops (*n* = 47) conducted. Adapting the #chatsafe guidelines in Australia to thirty-eight countries was studied.	Minimal adaptation was required to achieve a broader virtual international audience. It demonstrated the need for youth to feel better prepared to discuss suicide online.
Yeo, 2020 [33]	Qualitative narrative analysis	In Hong Kong, 136 anonymous personal Facebook stories of students on self-harm behaviours or suicidal thoughts, revealing hidden grievances and struggles, were analysed.	Users on social media platforms can post anonymously about their distress, giving them a buffer from the invalidation they face in real life and reducing the communication gap and interpersonal communication breakdowns.
Teo et al., 2020 [34]	Qualitative study	In the USA, thirty veterans were recruited through Facebook, and semi-structured interviews were held on responding to friends’ emotional distress on social media and potential interventions.	Many veterans showed hesitancy when posting about their troubles on Facebook but were willing to reach out to help other peers. Training veterans in suicide prevention could prove vital to their competence in responding to these distressing posts.
Robinson et al., 2015 [35]	Qualitative study	In Australia, ten researchers, 13 organisations, and 64 users were consulted on the nature and risks of Facebook use.	Social media allows users to talk to others with similar issues and provide emotional support. The benefits of social media suicide prevention initiatives far outweigh any risk associated.
Haines-Delmont et al., 2020 [36]	Cross-sectional study	The study applied machine learning in an acute mental health setting in the USA for suicide risk prediction through a smartphone app linked to Facebook and other platforms. It collected information about sleep, mood, step frequency, count, and engagement patterns with the phone.	An algorithm to assess suicide risk in inpatients can be created to provide crisis response to hesitant users to seek immediate help. This could be fully automated and independent of clinical judgement, but the legal and ethical implications must be considered.
Sumner et al., 2020 [37]	Analytical cross-sectional study	The study analysed 664 suicide-related articles on Facebook and adherence to suicide-reporting guidelines.	Assessment of Facebook data revealed that harmful components were frequently seen in news articles about suicide while protective elements were rare. Closer adherence to safe-reporting practices was associated with more shares.
Nathan and Nathan, 2020 [38]	Cross-sectional study	In the USA, a brief survey on the perception of suicide on social media platforms was completed, and 152 responded. The study assessed participants’ perceptions of suicide-related material on Facebook and associated factors such as gender.	A majority believed suicide was preventable and stated that the media glorified suicide. The females reported finding support on social media, while the males did not. Many did not believe suicide was stigmatised.
Lee et al., 2020 [39]	Cross-sectional study	In Australia, 7487 adults completed a survey on suicide literacy, stigma, and risk and participants were recruited via Facebook advertisements.	Facebook demonstrated efficacy and cost-efficiency in recruiting for a suicide prevention community. More extensive research is needed to make this sample more representative.
Brailovskaia et al., 2020 [40]	Cross-sectional study	An investigation of the possible mediating role of Positive Mental Health (PMH) for the relationship between Facebook addiction disorder and suicide-related outcomes among 209 cases from Germany was conducted.	The addictive Facebook use is related to suicide outcomes. PMH reduces this risk. It is recommended to consider addictive Facebook use and PMH in suicide risk assessment.
Tao and Jacobs, 2019 [41]	Secondary analysis	The study utilised an existing dataset of Facebook commentary, and there were 445 comments made on 156 anonymous posts.	Most of the participants support those sharing their experiences with depression and suicide. Therefore, social communication anonymously on social media can provide psychological support
Rashid Soron, T, 2019 [19]	Opinion article	The use of Facebook has increased in Bangladesh and has focused on preventing suicides.	Facebook could be used for developing a suicide database in Bangladesh and would be helpful in future policymaking. Ethical issues should be considered.

## Data Availability

The data are available upon request by Authors Sheikh Shoib and Sarya Swed.

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
