# Peer review of "Facebook and Suicidal Behaviour: User Experiences of Suicide Notes, Live-Streaming, Grieving and Preventive Strategies—A Scoping Review"

_ijerph, 2022, doi:10.3390/ijerph192013001_

Round 1

Reviewer 1 Report

Brief summary:

This article focuses on an important topic : the relationship between social media (Facebook) and suicidal behaviour. With this scoping review, the authors aim to explore the various aspects of suicidal behaviours associated with Facebook. The article is well written and encompasses important articles in this field to yield and discuss different aspects of social media – related suicidal behaviours. Please see below my comments.  

General concept comments

Introduction:

Lines 54-58: The paragraph on suicide is very succinct. While conveying important information on the topic, it would be interesting to add relevant information on the topic of suicide such as : risk factors, protective factors. This would strengthen even more the relevancy of your article considering the demographics of people using social media and people who are at risk of committing suicide.

Line 58 : This line could benefit from adding an example : what are the various cases?

Line 62: A reference is missing to acknowledge that this is indeed a new trend.

Line 80-84: The aim of this study could be clarified considering that ‘’Exploring’’ is very vast. A suggestion would be  to state clearly the intent of this review such as : This study aims to identify the various aspects of … . An hypothesis of what the aspects are could also be included. This would account for clarity.

Overall the introduction is clear and a bit succinct. However, the authors are encompassing the important literature on the subject and the paragraphs are ordered in an appropriate way. A paragraph on prediction of suicidal behaviours on social media (or 1-2 sentences) could be added considering the array of literature on the subject and the conclusion of this manuscript.

Methods:

The methods section would overall benefit from a structured reporting of the strategy employed to conduct the presented study. The use of guidelines such as the PRISMA guidelines should be used (ex.: https://prisma-statement.org/documents/PRISMA_2020_checklist.pdf) .

Figure 1 is a result of the search conducted and should not be part of the methods section. Please refer to Results reporting (https://prisma-statement.org/documents/PRISMA_2020_checklist.pdf).

Line 93: Please describe briefly the ‘’group of consulting psychiatrists’’.

The eligibility criterions for the identification of the studies are missing. Inclusions and exclusion criterions should be fully described.

The strategy for data collection and reporting should be indicated.

The full search strategies should be accessible to the readers. It is suggested that the strategies are added as supplemental material.

This scoping review only uses three databases (PubMed, Scopus and Google Scholar). Major databases such as PsycINFO or Web of Science are not explored. Considering the importance of the subject and the articles published on similar topics, the manuscript would benefit from including these other major databases.

Overall the Methods section would need significant improvement as the methodology is currently very opaque to the readership and it would be difficult to reproduce it with the current details provided by the authors. Eligibility criterions for to select the studies are not presented, and it is therefore not possible to derive how the studies were selected and why these particular studies were analyzed.

Results:

The lines 104-113 belongs to the Methods section if this was protocoled prior to the search. If not, it should mention that these are the aspects that were identified in the reported studies.

The reported key points should be defined. What consists of a key point and why? This could be explicated in the Methods section (data reporting). Otherwise, it will be difficult to derive conclusions from the heterogeneous key points collected.

In the different tables, Study details should be ‘’Article type’’ to be consistent with data reporting for scoping reviews.

The Report columns of the different tables should be divided. It is suggested to include a column: Population (to identify the study’s population) and a column about the type of data reported.

Overall the results section could be improved to further clarify the data collected. The present section is limited to the explicated Methodology.

Discussion:

Lines 133-134: This sentence belongs to the introduction.

Lines 133-137: It would be pertinent to add a sentence to highlight the amount of identified articles in your review prior to listing the different aspects of suicidal behaviours on social media.

Lines 155-157: The data presented is interesting. Sources should be cited.

The articles identified are not discussed in the Discussion section. Only the four themes are used across the discussion. It would be interesting to bridge the articles identified with the literature on the subject.

Limitations of the presented scoping review are only about the identified articles. It would be interesting to add the limitations of scoping reviews and the limitations due to the presented methodology.

The Future Clinical Implications section should be aligned with the articles identified in the scoping review. It would be suggested to conclude by adding a brief summary of ‘’what was done in this study’’ and ‘’what was found’’.

Minor comments:

Please verify the tense of the manuscript. For example, Line 63 : It is observed… while Line 70: It has been observed… .

Review:

Overall, this study tackles an important question in the fields of psychology and the use of social media. It is unclear at this point the exact methodology used to conduct this review and how data was collected and it is therefore difficult to make recommendations as to the identified articles.  A firm revision of the methodology would benefit the manuscript and increase significantly the quality of this study.

Author Response

Response to reviewers

We thank the editors and reviewers for their valuable time and helpful suggestions.

Reviewer #1

This article focuses on an important topic: the relationship between social media (Facebook) and suicidal behaviour. With this scoping review, the authors aim to explore the various aspects of suicidal behaviours associated with Facebook. The article is well written and encompasses important articles in this field to yield and discuss different aspects of social media–related suicidal behaviours. Please see below my comments. 

Thank you.

General concept comments

Introduction:

Lines 54-58: The paragraph on suicide is very succinct. While conveying important information on the topic, it would be interesting to add relevant information on the topic of suicide such as: risk factors, and protective factors. This would strengthen even more the relevancy of your article considering the demographics of people using social media and people who are at risk of committing suicide.

We have added risk factors and protective factors for suicides to upgrade the paragraph.

Line 58: This line could benefit from adding an example: what are the various cases?

We have updated the sentence.

Line 62: A reference is missing to acknowledge that this is indeed a new trend.

A reference has been added.

Line 80-84: The aim of this study could be clarified considering that ‘’Exploring’’ is very vast. A suggestion would be to clearly state the intent of this review such as This study aims to identify the various aspects of … . A hypothesis of what the aspects are could also be included. This would account for clarity.

The statement has been updated.

Overall, the introduction is clear and a bit succinct. However, the authors encompass the important literature on the subject, and the paragraphs are ordered in an appropriate way. A paragraph on the prediction of suicidal behaviours on social media (or 1-2 sentences) could be added considering the array of literature on the subject and the conclusion of this manuscript.

A brief sentence has been added about AI methods.

Methods:

The methods section would overall benefit from a structured reporting of the strategy employed to conduct the presented study. The use of guidelines such as the PRISMA guidelines should be used (ex.: https://prisma-statement.org/documents/PRISMA_2020_checklist.pdf).

The study selection process is shown according to PRISMA guidelines.

Figure 1 is a result of the search conducted and should not be part of the methods section. Please refer to the Results reporting (https://prisma-statement.org/documents/PRISMA_2020_checklist.pdf).

Figure 1 has been moved to results.

Line 93: Please describe briefly the ‘’group of consulting psychiatrists’’.

The group comprised consultant psychiatrists, who are medical specialists in mental health, as the aim of the study is related to psychiatry and mental health. It is not ‘consulting psychiatrists’.

The eligibility criteria for the identification of the studies are missing. Inclusions and exclusion criteria should be fully described.

This is a scoping review, and adequate information has been provided.

“Articles related to announcing suicide on Facebook, live streaming suicidal behaviour, grieving suicide, and preventing suicides”

The strategy for data collection and reporting should be indicated.

This is a scoping review, and adequate information has been provided.

The full search strategies should be accessible to the readers. It is suggested that the strategies are added as supplemental material.

We want to emphasise that this is a scoping review, not a systemic one. Experts selected the studies. The search strategies are provided in the methods.

This scoping review only uses three databases (PubMed, Scopus and Google Scholar). Major databases such as PsycINFO or Web of Science are not explored. Considering the importance of the subject and the articles published on similar topics, the manuscript would benefit from including these other major databases.

The review is focused on selected themes, and adequate studies have been found. Many institutions in low- and middle-income countries do not have access to PsycINFO or Web of Science.

Overall, the Methods section would need significant improvement as the methodology is currently very opaque to the readership, and it would be difficult to reproduce it with the current details provided by the authors. Eligibility criterions for to select the studies are not presented, and it is therefore not possible to derive how the studies were selected and why these particular studies were analysed.

Thanks for your suggestion, we want to emphasise that this is a scoping review, not a systemic one. Therefore we try to make an outlook  about facebook and suicide behavior and we don't do statistical analysis. However we have improved this part.

Results:

Lines 104-113 belong to the Methods section if this was protocoled prior to the search. If not, it should mention that these are the aspects that were identified in the reported studies.

The themes targeted are mentioned in the method section. The first paragraph of the results indicates the tables containing information.

The reported key points should be defined. What consists of a key point and why? This could be explicated in the Methods section (data reporting). Otherwise, it will be difficult to derive conclusions from the heterogeneous key points collected.

The key points are defined in the methods.

In the different tables, Study details should be ‘’Article type’’ to be consistent with data reporting for scoping reviews.

The column topic has been changed.

The Report columns of the different tables should be divided. It is suggested to include a column: Population (to identify the study’s population) and a column about the type of data reported.

Thank you for the suggestion. The initial versions of the manuscript had information shown in that way. However, it led to crowded tables and unpleasing reading.

Overall, the results section could be improved to further clarify the data collected. The present section is limited to the explicated Methodology.

Thank you for the valuable suggestion. Most of the data found are observational and would lead to a repetition of information in results and discussion.

Discussion:

Lines 133-134: This sentence belongs to the introduction.

We have edited this section and removed this sentence.

Lines 133-137: It would be pertinent to add a sentence to highlight the amount of identified articles in your review prior to listing the different aspects of suicidal behaviours on social media.

We have updated this suggestion to include at the start of the discussion.

Lines 155-157: The data presented is interesting. Sources should be cited.

The references have been added, and the section has been updated.

The articles identified are not discussed in the Discussion section. Only the four themes are used across the discussion. It would be interesting to bridge the articles identified with the literature on the subject.

Thank you, and we have added the references to the discussion.

Limitations of the presented scoping review are only about the identified articles. It would be interesting to add the limitations of scoping reviews and the limitations due to the presented methodology.

A statement was added about the limitations of the methodology.

The Future Clinical Implications section should be aligned with the articles identified in the scoping review. It would be suggested to conclude by adding a summary of ‘’what was done in this study’’ and ‘’what was found’’.

Thank you, and we have added statements to update the section.

Minor comments:

Please verify the tense of the manuscript. For example, Line 63: It is observed… while Line 70: It has been observed.

We have corrected specific writing errors.

Review:

Overall, this study tackles an important question in the fields of psychology and the use of social media. It is unclear at this point the exact methodology used to conduct this review and how data was collected, and it is therefore difficult to make recommendations as to the identified articles.  A firm revision of the methodology would benefit the manuscript and significantly increase the quality of this study.

Thank you for your valuable suggestions, and we have updated the methods section.

Reviewer 2 Report

The manuscript „Facebook and suicidal behaviour: user experiences of suicide notes, live-streaming, grieving and preventive strategies. A scoping review.“ examines how Facebook and various aspects of suicidal behavior are associated. In the end the authors introduce 28 studies, summarize their key points, and conclude with recommendations. The manuscript is well written and technically sound. However, the numbers of studies the authors found is quite small and some of the conclusions should be weakened. There are a few issues I would like to address.

Abstract

-          It does not become clear what the authors mean that a group of consultant psychiatrists screened the records and extracted relevant data

-          The number of studies included in the review is missing, could the authors add this number?

Methods

-          Are the authors the group of consultant psychiatrists? Please clarify.

Results

-          The key points are partly far-fetched given that the studies are mostly case studies.

Discussion

-          The authors cannot be made responsible for the studies they found but the discussion needs to put more emphasize on lack of studies and the quality of the found studies. A lot of the studies were only case reports and qualitative (the missing of quantitative data is indeed mentioned by the authors). Given the little evidence and the case studies, I am not sure whether the authors should draw such big conclusions except for the fact that there is barely any research on this topic, the same applies to the recommendations. I agree with the authors that there is more work to be done in this specific area.

-          I miss the reference to more “modern” social platforms such as Instagram or even TikTok that also have come to the fore on social media research.

Minors

-          There is a word missing on line 39 “harmful since negatively affects audience”

-          There are multiple sentences in the manuscript that are connected with “and” but might rather “stand alone” because some sentences are quite long and it is hard to follow.

-          Please clarify authors’ contribution. Who collected data, who wrote the manuscript and so on…

Author Response

Response to reviewers

We thank the editors and reviewers for their valuable time and helpful suggestions.

Reviewer #2

The manuscript “Facebook and suicidal behaviour: user experiences of suicide notes, live streaming, grieving and preventive strategies. A scoping review.“ examines how Facebook and various aspects of suicidal behaviour are associated. In the end, the authors introduce 28 studies, summarize their key points, and conclude with recommendations. The manuscript is well-written and technically sound. However, the number of studies the authors found is quite small, and some of the conclusions should be weakened. There are a few issues I would like to address.

Thank you, and we have mentioned these as limitations.  

Abstract

-          It does not become clear what the authors mean that a group of consultant psychiatrists screened the records and extracted relevant data

We have updated the methods section to clarify this.

-          The number of studies included in the review is missing; could the authors add this number?

We have included the number in the abstract now.

Methods

-          Are the authors the group of consultant psychiatrists? Please clarify.

Yes, that is correct. Now mentioned on lines 100-102.

Results

-          The key points are partly far-fetched given that the studies are mostly case studies.

We have mentioned that the panel of experts decided on the key points. The scoping review aims to emphasise essential aspects when there are not many relevant publications.

Discussion

-          The authors cannot be made responsible for the studies they found but the discussion needs to put more emphasis on the lack of studies and the quality of the found studies. A lot of the studies were only case reports and qualitative (the missing quantitative data is indeed mentioned by the authors). Given the little evidence and the case studies, I am not sure whether the authors should draw such big conclusions except for the fact that there is barely any research on this topic; the same applies to the recommendations. I agree with the authors that there is more work to be done in this specific area.

Thank you, and we agree. We have modified the discussion accordingly. However, we feel the recommendations are now appropriate until further research is available. 

-          I miss the reference to more “modern” social platforms such as Instagram or even TikTok that also have come to the fore in social media research.

We agree that these modern social media may be more relevant for younger people. However, our manuscript solely focuses on Facebook.

Minors

-          There is a word missing on line 39 “harmful since negatively affects audience”

We have corrected the error.

-          There are multiple sentences in the manuscript that are connected with “and” but might rather “stand-alone” because some sentences are quite long and it is hard to follow.

We have separated some of the sentences with ‘and’.

-          Please clarify the authors’ contribution. Who collected data, who wrote the manuscript and so on…

Sheikh Shoib: conceptualization, writing first draft

Sheikh Shoib, Tan weilling Amanda: collecting data

All other authors have contributed to the present paper equally in writing first draft and revising final version.

Round 2

Reviewer 1 Report

The authors responded adequately to my comments.